# Echocardiographic Assessment of Patients with Glycogen Storage Disease in a Single Center

**DOI:** 10.3390/ijerph20032191

**Published:** 2023-01-25

**Authors:** Jaehee Seol, Seyong Jung, Hong Koh, Jowon Jung, Yunkoo Kang

**Affiliations:** 1Department of Pediatrics, Yonsei University Wonju College of Medicine, Wonju 26426, Republic of Korea; 2Division of Pediatric Cardiology, Department of Pediatrics, Yonsei University College of Medicine, Seoul 03722, Republic of Korea; 3Division of Gastroenterology, Hepatology, and Nutrition, Department of Pediatrics, Severance Children’s Hospital, Severance Pediatric Liver Disease Research Group, Yonsei University College of Medicine, Seoul 03722, Republic of Korea

**Keywords:** glycogen storage disease, cardiac function, left ventricle mass, left ventricle thickness, global longitudinal strain

## Abstract

Glycogen storage disease (GSD) is a hereditary metabolic disorder caused by enzyme deficiency resulting in glycogen accumulation in the liver, muscle, heart, or kidney. GSD types II, III, IV, and IX are associated with cardiac involvement. However, cardiac manifestation in other GSD types is unclear. This study aimed to describe whether energy deprivation and the toxic effects of accumulated glycogen affect the heart of patients with GSD. We evaluated the left ventricle (LV) wall mass, LV systolic and diastolic function and myocardial strain with conventional echocardiography and two-dimensional speckle-tracking echocardiography (2D STE) in 62 patients with GSD type I, III, VI and IX who visited the Wonju Severance Hospital in 2021. Among the GSD patients, the echocardiographic parameters of 55 pediatrics were converted into z-scores and analyzed. Of the patients, 43 (62.3%), 7 (11.3%) and 12 (19.4%) patients were diagnosed with GSD type I, type III, and type IX, respectively. The median age was 9 years (range, 1–36 years), with 55 children under 18 years old and seven adults over 18 years. For the 55 pediatric patients, the echocardiographic parameters were converted into a z-score and analyzed. Multiple linear regression analysis showed that the BMI z-score (*p* = 0.022) and CK (*p* = 0.020) predicted increased LV mass z-score, regardless of GSD type. There was no difference in the diastolic and systolic functions according to myocardial thickness; however, 2D STE showed a negative correlation with the LV mass (r = −0.28, *p* = 0.041). Given that patients with GSD tend to be overweight, serial evaluation with echocardiography might be required for all types of GSD.

## 1. Introduction

Glycogen storage disease (GSD) consists of inherited inborn errors of metabolism caused by mutations in genes encoding enzymes involved in glucose metabolism [1]. GSD is classified according to enzyme deficiency, and there are more than 12 types. The overall incidence of GSD is estimated to be 1 in 25,000–43,000 live births [1,2]. Glycogen that cannot be broken down into glucose accumulates mostly in the liver, skeletal muscle, heart, and/or kidney. GSD is classified as hepatic or muscle glycogenosis according to the organ where glycogen is mainly accumulated. Hepatic GSD includes types I, III, IV, VI, IX, and GSD 0, and muscle GSD includes types V, VII, X, XI, XII, XIII, and XIV. Types II, III, IV, and IX affect both skeletal muscles and the heart. Disease manifestations are caused either by energy deficiency or the toxic effects of the accumulated glycogen. The deposit of glycogen induces organomegaly and impaired organ function and cellular structure [3]. Glycogen accumulation within the cells may result in toxicity causing cell dysfunction or death. Clinical features of hepatic GSD arise from the inability to mobilize glucose during fasting or exercise, which may lead to hepatomegaly, hypoglycemia, lactic acidosis, and hyperuricemia [2,4,5]. Prolonged hypoglycemia may cause damage to the central nervous system [6]. A characteristic feature of muscular-type GSD is muscle weakness (hypotonia); dyslipidemia and metabolic acidosis are also found on laboratory testing. In both types of GSD, short stature can occur, and liver enzymes can be elevated. Clinical severity and prognosis differ according to GSD type and age. In particular, types II, III, IV, and IX are known to be associated with cardiac involvement that causes a hypertrophic response [7,8,9,10]. Cardiomyopathy is not known to be associated with GSD type 1, though pulmonary hypertension has been reported. [11]

The metabolic diseases, such as diabetes mellitus and obesity, can cause functional or structural myocardial disease via metabolic derangement [10,12]. Hyperglycemia and/or accumulation of free fatty acids and triglycerides (TGs) may contribute as pro-inflammatory factors and cause myocardial toxicity [13]. This study sought to establish the energetic deficiency or toxic effects of acidosis or dyslipidemia that can cause cardiac injury or myocardial hypertrophic change in patients with GSD, including type I GSD, which is most common in Korea. The study aimed to evaluate echocardiographic assessments of general and cardiac manifestations in patients with GSD.

## 2. Materials and Methods

### 2.1. Study Population

This was a single-center retrospective cohort study. The study comprised 64 patients diagnosed with GSD using next generation sequencing and liver biopsy, who visited the Wonju Severance Hospital between March 2021 and April 2022. Patients were grouped by GSD type, and two patients (one with GSD type VI and one with an unknown type) were excluded as they could not be compared. The patient’s GSD type, age, height, weight, sex, and laboratory findings were recorded at the time of echocardiographic assessments. Body mass index (BMI) was calculated as weight (kilograms) divided by height (meters) squared, and then for children under the age of 18 years, the BMI z-score was calculated using the Centers for Disease Control and Prevention (CDC) growth charts. According to CDC criteria a z-score ≥ 1.45 (85 percentile) was defined as overweight, and a z-score ≥ 2 was defined as obesity.

### 2.2. Echocardiography Analysis

A single experienced pediatric echocardiographer performed transthoracic echocardiography using a Philips EPIQ 5c system (Philips Medical System Andover, Andover, MA, USA). Myocardial mass and left ventricle (LV) systolic function were measured using the M-mode at short axis view. Doppler signal quality was enhanced by lowering the Nyquist limit to 10–30 cm/s, using the lowest wall filter setting with minimal optimal gain, decreasing Doppler sample volume size to 5 mm, and optimizing the sweep speed to at least 100 mm/s.

LV mass was calculated using the LV wall thickness and the LV cavity diameter measured during end diastole. For optimal LV mass indexing, according to previous studies, younger children aged < 8 years and children and adolescents aged <19 years were indexed by dividing the LV mass by height (meters) increased to a power of 2.0 and 2.7, respectively. Adults aged >19 years were indexed by dividing the LV mass by body surface area (BSA) [14,15,16]. The normal ranges for adult male and female patients were 49–115 g/m^2^ and 43–95 g/m^2^, respectively [17]. The LV systolic function was evaluated with ejection fraction (EF) and fractional shortening (FS) by measuring M-mode at the short axis view. We defined normal EF as ≥55% and FS as >28% in both children and adults [18,19]. The LV diastolic function was assessed using the E/A ratio and E/e’ ratio by measuring the pulse-wave tissue Doppler. Peak velocities of the early filling (E) and late filling (A) waves and the E/A ratio of peak velocities were measured using trans-mitral inflow velocities. Early diastolic mitral tissue velocities at the septal and lateral mitral annulus (e’ velocity) were obtained using tissue Doppler imaging, and the E/e’ ratios were calculated. The LV diastolic dysfunction was identified when the E/A ratio was reversed (E/A ratio < 1) or both E/A and E/e’ z-scores were outside the −2–2 range. The right ventricle (RV) systolic function was assessed using tissue Doppler-derived peak systolic velocity at the RV free wall (RV S’). Two-dimensional speckle-tracking echocardiography (2D STE) with apical four- and two-chamber long axis views was used to identify the LV global longitudinal strain (GLS) via manual tracing of the endocardial contour at end-systole. The following frames were automatically analyzed by temporal tracking of acoustic speckles. Longitudinal strains for each segment were measured and expressed as a bull’s eye.

The echocardiographic evaluation is confounded by the effect of age, body weight, and height. All echocardiographic parameters, which have different normal reference values according to age, were converted into z-scores and analyzed for comparison. The BSA-adjusted z-score described by Dallaire et al. was used for left-sided pulse-wave Doppler and tissue Doppler imaging [20]. The RV systolic function using tricuspid annular peak systolic velocity was transformed into age-adjusted z-scores as described by Koestenberger et al. [21]. According to Foster et al., the LV mass was converted to height adjusted z-scores in pediatric patients (<19 years) [14]. The LV GLS was transformed to BSA-adjusted z-scores as described by Dallaire et al. A z-score value between −2 and 2 was considered normal [22].

### 2.3. Statistical Analysis

In pediatric patients, the echocardiographic parameters were compared with the z-sore adjusted for age, weight, and height, and seven adult patients were excluded from the analysis. All statistical analyses were conducted using the Statistical Package for the Social Sciences, version 25.0 (IBM Corp., Armonk, NY, USA). For continuous variables, medians with interquartile ranges (IQRs) were calculated and compared using analysis of variance. (ANOVA) was used to analyze the difference between the three groups, and post-hoc Bonferroni analysis was used for subsequent analysis. Pearson’s correlation for parametric data was used to assess the relationship between the BMI z-score, LV mass, heart function, and laboratory findings. Significant variables were further analyzed using the stepwise methods of multiple linear regression analysis to isolate predictors of an increase in LV mass. All tests were two tailed, and statistical significance was set at *p* < 0.05.

## 3. Results

Sixty-four patients with GSD visited our institution during the study period. One patient with GSD type VI and another one with GSD of unknown type were excluded from the analysis. Thus, 62 patients were included in the study. The clinical and demographic characteristics of the patients are listed in Table 1. Of the total patients (43 males and 19 females), 43 (62.3%), 7 (11.3%), and 12 (19.4%) were diagnosed with GSD types I, III, and IX, respectively. The median age at presentation was 9 years (range, 1–36 years): 55 children (age, <18 years) and seven adults (age, >18 years). The mean BMI was 19.6 (range, 13.1–26.5). There was no statistical difference in the distribution of age groups between GSD types. In laboratory findings, the mean levels of uric acid (reference level, <6.0 mg/dL) and total cholesterol (reference level, <170 mg/dL) were increased. In addition, the median level of lactate (reference level, 0.5–2.0 mmol/L) was also increased. The median values for TGs (reference level, <200 mg/dL), aspartate transaminase (AST) (reference level, <40 U/L), alanine transaminase (ALT) (reference level, <40 U/L), and creatine kinase (CK) (reference level, <170 U/L) were within the normal range. Uric acid and lactate were significantly elevated in patients with GSD type I compared with those in patients with other GSD types. The AST, ALT, and CK levels were increased in patients with GSD type III. Total cholesterol and TG levels were higher in children with GSD types I and III compared with those in children with type IX. All patients had their blood pressure measured on the day of echocardiography and blood tests, and all blood pressures were within the normal range. There were three patients with congenital heart disease discovered incidentally by echocardiography in this study. These patients were diagnosed with ventricular septal defect, atrial septal defect, and patent ductus arteriosus, respectively. However, these were not hemodynamically significant defects. There was also no reported cardiovascular symptom in these patients. None of our patients had pulmonary hypertension.

The comparison of BMI and echocardiographic parameters adjusted for the age, height, and weight of pediatric patients is listed in Table 2. The median BMI z-score was 1.2 (range, −2.5–2.24), and there was no difference in the BMI z-score according to GSD type. However, of the five patients with BMI z-scores >2, 2, 2, and 1 had GSD types I, III, and IX, respectively. The mean LV mass indexed z-score was 0.3 (range, −1.05–3.56). When comparing myocardial thickness between GSD types, the LV mass z-score (*p* =0.037), interventricular septum diameter (IVSd) z-score (*p* < 0.001), and LV posterior wall thickness diameter (LVPWd) z-score (*p* < 0.001) were relatively high in the patients with GSD type III. In this study, most of the pediatric patients had normal LV mass z-scores, except for two patients (GSD types I and III) with an LV mass z-score >2. One patient with a normal LV mass z-score had an IVSD z-score of >2 and had GSD type I. Analysis of diastolic and systolic function showed no significant differences between the GSD types. Four patients had an E/A z-score less than −2 and 15 had an E/e’ z-score greater than 2 (Figure 1). However, there were no pediatric patients with both E/A and E/e’ outside the −2–2 range or reversal of E/A (ratio <1). The RV s’, indicating RV function were all within the normal range, as evaluated by z-score. All values of EF and FS remained within normal ranges. There were no patients with decreased LV myocardial strain as evaluated by z-score, and there was no difference between GSD types. Similar to the results in Table 1, there was a difference in laboratory findings between groups.

Pearson correlation analysis showed that LV mass index z-scores were independent of age, gender, and laboratory findings, such as uric acid, lactate, total cholesterol, triglyceride, AST, and ALT. However, the BMI z-score (*p* = 0.005), HDL (*p* = 0.036) and CK (*p* = 0.005) correlated with the LV mass z-score in children (Table 3). Multiple linear regression analysis showed that the BMI z-score (*p* = 0.022) and CK (*p* = 0.020) predicted an increased LV mass z-score, regardless of GSD type (Table 4).

EF and FS for the LV systolic function evaluation and tissue Doppler for diastolic function evaluation all remained within normal ranges. There were no significant changes in the LV systolic function or diastolic function with increasing LV mass. However, 2D STE showed a negative correlation between GLS and the LV mass: the average GLS decreased as the LV mass increased (Figure 2). In the correlation analysis between the patients’ characteristics and changes in cardiac function, there was a significant negative correlation between the BMI z-score and E/A z-score, and positive correlation between TG increase and the E/e’ z-score (Appendix A).

Table 5 shows the echocardiographic features of the seven adult patients with GSD types I (n = 5) and III (n = 2). Their ages ranged from 20 to 36 years (four males and three females). All male patients had LV mass indexes in the normal range; however, one female patient with GSD type III had a slightly elevated LV mass index. No. 5 was a 34-year-old male patient diagnosed with GSD type I. His LV mass index (109 g/m^2^) was slightly increased without obesity, and his E/A ratio (0.78) was revered. EF was preserved, but the average GLS showed a relatively low value of −17.5% (Figure 3). The laboratory findings of this patient showed an increase in lactate (3.49 mmol/L) and total cholesterol (348 mg/dL). No. 6 was a 25-year-old female patient diagnosed with GSD type III. LV mass index (100.7 g/m^2^) was increased above the normal range with a BMI of 19.3. In addition, she showed diastolic dysfunction indicated by E/A (2.34) and E/e’ (8.73). RV function, EF, FS, and strain remained within normal ranges. In laboratory findings, AST and ALT were slightly increased.

## 4. Discussion

The heart has complex metabolic processes to use energy to maintain its function. If this metabolic process is disrupted, heart function or structure is affected. The exact mechanisms of cardiac manifestations are currently unknown; however, they seem to correlate with the accumulation of by-products, toxic effects, and lack of energy [12,23,24,25]. Cardiac involvement in terms of functional and structural changes in many metabolic diseases has been reported. GSD is an inherited metabolic diseases of glycogen metabolism. Although GSD severity varies with type, it is usually associated with hypoglycemia, hyperuricemia, hyperlactatemia, and dyslipidemia. Clinical manifestation depends on the main organ where non-degraded glycogen is deposited. GSD types II, III, IV, and IX are known to be associated with cardiac involvement. Glycogen accumulation in the myocardium leads to myocardial injury, myocardial hypertrophy, and conduction disorder. GSD types II and III (known for LV hypertrophy due to the deposition of glycogen in the myocardium) and IX (cardiac kinase deficiency) are known to cause heart failure in young infants. Previous studies have mainly focused on the LV mass evaluation and heart function limited to GSD types II and III. However, studies on the overall cardiac evaluation of patients with GSD are lacking [9,26,27]. This study sought to use standard echocardiography and 2D STE to determine whether the energy deficiency, acidosis, dyslipidemia, or obesity seen in patients with GSD resulted in structural and functional changes of the heart, as seen with other metabolic diseases such as diabetes mellitus or dyslipidemia.

In this study, 62 patients were examined, including children and adults. Pediatric patients (age < 18 years) accounted for 76% of all patients, and there was a greater number of male (62%) than female patients. These findings were similar to the results of the GSD prevalence study based on the Health Insurance Review and Assessment Servis (HIRA) research in Korea [28]. It was difficult to compare BMI according to types because the age and gender distribution of each patient group was different. As expected, abnormalities in laboratory findings differed depending on GSD type; however in the case of type IX, laboratory findings, including total cholesterol and TG, were relatively closer to the normal range than those in patients with other GSD types.

In the analysis comparing echocardiography parameters of cardiac function including tissue Doppler, EF and FS did not differ with GSD type. As previously reported, type III showed a tendency of thicker myocardium compared to that in patients with other GSD types. However, two patients with GSD type I, including one child and one adult, showed increased LV mass above the normal range. This suggests that, besides the characteristics of the GSD type, there may be other factors that affect the heart. These patients showed hyperuricemia, lactic acidosis, dyslipidemia, and increased AST and ALT due to poor metabolic control.

The LV mass index z-score was independent of age, gender, and laboratory findings, except for CK. In our study, the BMI z-score and CK showed a significant association with LV hypertrophy in GSD. As the age-adjusted BMI increased, the LV mass and LV wall thickness also increased. This finding was similar to those of previous studies that showed an association between obesity and LV hypertrophy in the general population [29]. Patients with GSD need regular meals during the day and night to prevent hypoglycemia. This commonly causes obesity in these patients. These results imply the need for exercise to manage obesity in patients with GSD. Cardiac evaluation should also be performed regularly in patients with GSD who are overweight as in the general population with obesity. CK is a biomarker that is increased during muscle damage, and it is increased in GSD types II, III, and V, which are mainly related to muscle involvement. In our study, there were some patients with CK levels higher than normal in patients with GSD types I and IX [30]. Although not implemented in our study, it would be better to compare CK-MB and evaluate echocardiography to clarify the relevance of CK as a clinical biomarker for LV hypertrophy.

In children, although there were patients with LV mass index z-scores above the normal range, systolic and diastolic heart function remained in the normal range. There was no difference in LV function according to LV mass change, which is inconsistent with the findings of previous studies suggesting a decrease in LV diastolic function associated with LV hypertrophy in children with obesity in the general population [25,29]. This may be because there were only two pediatric patients with an LV mass z-score of >2; thus, it was not a comparative evaluation of patients with LV hypertrophy and those without. A recent meta-analysis by Burden et al. showed reduced diastolic function with an increased E/e’ ratio and decreased E/A ratio in children with obesity [31]. This is similar to the results of our study showing that a higher BMI z-score was associated with a lower E/A z-score (r = −0.286, *p* = 0.038), and a higher TG was associated with a higher E/e’ z-score (r = −0.284, *p* = 0.038). Considering that the proportion of patients aged less than 10 years was higher in our study than in other studies that evaluated diastolic function in children with obesity, serial echocardiography evaluation is necessary for assessing functional changes in the heart as these pediatric patients mature. As a measurement to evaluate systolic function, EF and FS were maintained within normal ranges and were not related to myocardial thickness. However, the strain z-score decreased as the LV mass increased. In general, EF is useful for more easily evaluating LV systolic function but has limitations in predicting functional capacity and prognosis [32,33]. 2D STE is a quantitative measurement method for myocardial deformity and can detect impairment of subclinical LV systolic function in asymptomatic patients [34,35,36]. Our analysis suggests that changes in myocardial thickness in patients with GSD may be accompanied by functional changes and a need for regular strain evaluation.

Among adults, there were two patients with LV diastolic dysfunction and a slight increase in LV mass index. Neither of them had any symptoms or cardiovascular disease, including hypertension. The single male adult patient with GSD type I showed diastolic dysfunction and reduced strain despite a relatively young age of 25 years. His disorder had not been properly managed for a long time, and it is possible that he had cardiac injury due to metabolic abnormality. This suggests that echocardiography is needed to evaluate cardiac involvement in all types of GSD, not only in the types already known to be associated with cardiac involvement.

Numerous studies have evaluated cardiac function in GSD types II and III. However, to our knowledge, this is the first study to use standard echocardiography and 2D STE in patients with GSD types I, III, and IX. Our study is meaningful in that it assessed the cardiac evaluation of a relatively large number (n = 64) of patients with GSD in a single center in Korea. It is already known that cardiomyopathy occurs due to cardiac deposition of glycogen in some GSD types. In this study, there was no significant myocardial injury in patients with GSD according to laboratory findings and increased BMI. However, our study showed subclinical functional change in the heart according to myocardial thickness with 2D STE or hypertrophy and GSD type. The associations between BMI and E/A and TG level and E/e’ in our results suggest the need for further research and follow-up on functional changes using echocardiography in patients with GSD, especially those who are overweight. LV diastolic dysfunction could precede LV systolic dysfunction and is known as a contributor to mortality from cardiovascular disease in people with obesity. GLS could also be an early marker for LV dysfunction. Regular screening with echocardiography, including wall thickness measurement and diastolic functional evaluation, and 2D STE allows for the early diagnosis of subclinical heart changes in patients with GSD.

This study had some limitations. First, there was no healthy control group. The study could not establish whether there are differences in heart structure and functional changes in patients with GSD compared with those in the general population. Second, there may have been selection bias due to the differences in the number of patients between types. Third, a one-off blood test has limitations in representing a patient’s previous metabolic status. However, the study is meaningful in that overall echocardiography was performed on a large number of patients with GSD, including GSD type I, and subtle cardiac structure changes in subclinical states were measured through 2D STE.

## 5. Conclusions

There is still no specific curative treatment for GSD. The management of GSD varies with GSD type, and lifelong management is required to control symptoms or avoid possible complications, such as liver failure and cardiomyopathy. Since metabolic disease is often related to heart dysfunction and structural change, our results showed the effects of metabolic aberration on the hearts of patients with GSD. In our study, metabolic abnormalities, such as hyperuricemia, lactic acidosis, and dyslipidemia, caused by GSD did not have a significant effect on the function of the heart structure; however, there was a significant change in myocardial thickness with weight gain. Given that patients with GSD tend to be overweight or obese, steady cardiac evaluation through echocardiography is required for all types of GSD patients, including types not known to be associated cardiac involvement. Not only the evaluation of myocardial thickness, but also the evaluation of diastolic function and 2D STE will enable early detection of subclinical cardiac dysfunction. Future studies with a larger number of adult patients will be able to observe the long-term effects of metabolic abnormalities on the heart of patients with GSD. In addition, it is necessary to find out whether diet, optimal metabolic control, and exercise management in patients with GSD have a positive effect on changes in cardiac structure and function through follow-up echocardiography.

## Figures and Tables

**Figure 1 ijerph-20-02191-f001:**
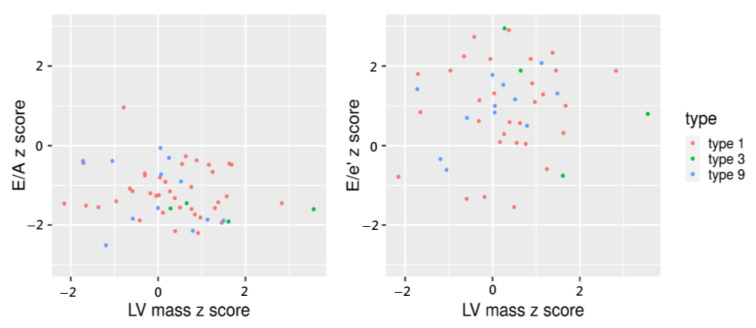
Tissue Doppler imaging for the evaluation of the left ventricle (LV) diastolic function.

**Figure 2 ijerph-20-02191-f002:**
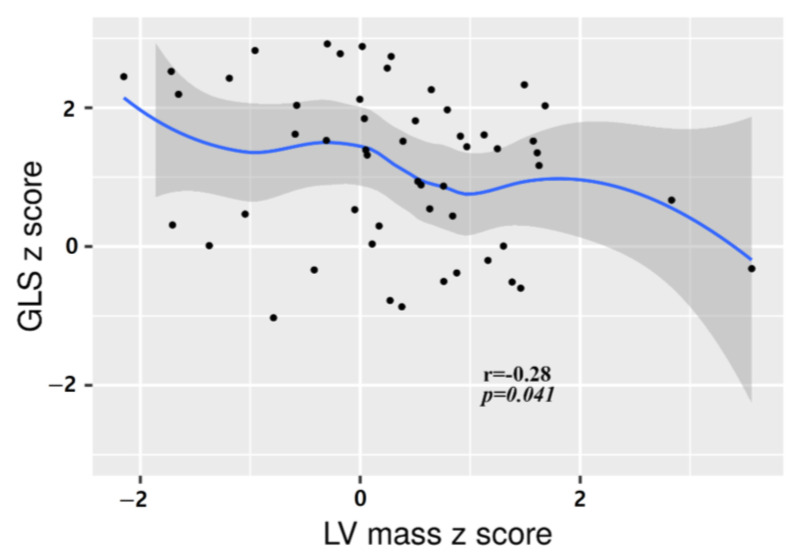
Correlation between global longitudinal strain (GLS) z-score and left ventricle mass z-score.

**Figure 3 ijerph-20-02191-f003:**
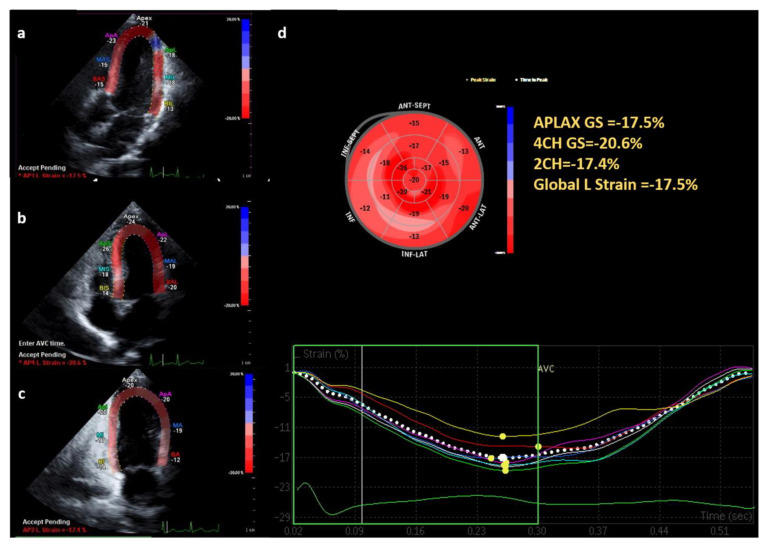
Left ventricular GLS assessment with two-dimensional speckle tracking echocardiography. The figure demonstrates analysis of left ventricular GLS from the three-chamber (**a**), four-chamber (4CH) (**b**) and two-chamber (2CH) (**c**) views. (**d**) Polar map with the regional values and the GLS value calculated from the 17 segments which is within the normal value.

**Table 1 ijerph-20-02191-t001:** Characteristics of the study population.

Variables	Total (%) N = 62 (100)	Group			*p*-Value
		Type 1 (%) N = 43 (62.3)	Type 3 (%) N = 7 (11.3)	Type 9 (%) N = 12 (19.4)	
Age (years)	9.0 (6.0, 14.0)	9.0 (6.5, 15.0)	5.0 (3.0, 20.0)	6.0 (4.5, 9.0)	0.051
0~7 y, n(%) 8~18 y, n(%) ≥19 y, n(%)	29 (46.8) 26 (41.9) 7 (11.3)	17 (39.5) 21 (48.8) 5 (11.6)	4 (57.1) 1 (14.2) 2 (28.6)	8 (66.7) 4 (33.3) 0	
Wt (kg)	30.0 (20.0, 50.0)	35.4 (23.0, 52.8)	20.0 (17.5, 56.0)	19.6 (15.6, 31.4)	
Hct (cm)	127.0 (110.0 150.0)	132.3(115.5, 132.3)	104.0(95.5, 163.5)	111.8(102.3, 126.5)	
BSA (m2)	1.1 (0.8, 1.4)	1.1 (0.8, 1.5)	0.8 (0.7, 1.3)	0.8 (0.7, 1.1)	
BMI	19.1 (17.4,21.0)	19.6 (17.5, 22.5)	18.9 (18.5, 20.8)	18.1 (15.4, 19.5)	
Gender					0.036
Male	43 (69.4)	27 (62.8)	4 (57.1)	12 (100.0)	
Female	19 (30.6)	16 (37.2)	3 (42.9)	0	
Uric acid (mg/dL)	6.3 (4.2, 8.1)	7.2 (5.8, 8.5)	4.2 (3.6, 6.2)^a^	3.3 (2.7, 3.8) ^a^	<0.001
Lactate (mmol/L)	2.2 (1.8, 3.0)	2.7 (2.0, 3.2)	1.7 (1.4, 2.2) ^a^	1.7 (1.3, 2.1) ^a^	<0.001
Total Col.(mg/dL)	192.5 (163.3, 231.5)	203.0 (176.0, 238.0) ^a^	190.0 (179.0, 257.5) ^a^	147.5 (135.8, 168.3)	0.001
HDL (mg/dL)	43.5 (38.3, 54.5)	43.0 (38.0, 50.0)	40.0 (36.5, 42.5)	53.5 (42.5, 64.5)	0.064
LDL (mg/dL)	95.0 (79.0, 126.0)	110.0 (85.8, 133.3) ^a^	107.0 (85.0, 119.0) ^a,b^	83.0 (62.5, 92.8) ^b^	0.019
TG (mg/dL)	185.0(118.3, 367.5)	233.0(139.0, 395.0) ^a^	242.0(148.0, 367.0) ^a^	89.5(65.5, 110.0)	<0.001
AST (U/L)	31.0 (25.0, 58.0)	30.0 (24.0, 38.0) ^a^	109.0 (60.0, 357.0)	33.0 (25.0, 35.3) ^a^	0.001
ALT (U/L)	25.5 (17.0, 67.0)	26.0 (18.0, 43.0)	254.0 (87.0, 605.0)	17.0 (15.0, 24.8)	<0.001
CK (U/L)	120.5 (91.0, 144.0)	117.0(68.0, 133.0) ^a^	402.0(117.0, 629.0) ^b^	130.0 (99.8, 182.8) ^a,b^	0.018

*p* values represent overall differences across groups as determined by one-way analysis of variance (ANOVA) or Kruskal-Wallis’ H-test for continuous variables. ^a^,^b^ Same letters indicate no statistical significance based on Tukey’s post-hoc analysis or Bonferroni Correction Method. AST, aspartate transaminase; ALT, alanine transaminase; BSA, body surface area; CK, creatinkinase; Hct, height; TG, triglyceride; Total Col., total cholesterol; Wt, weight.

**Table 2 ijerph-20-02191-t002:** BMI z scores and echocardiography results of children aged < 18 years.

Variables	Total (N = 55)	Group			*p*-Value
		Type 1 (N = 38)	Type 3 (N = 5)	Type 9 (N = 12)	
BMI (z)	1.2 (0.3, 1.6)	1.2 (0.4, 1.5)	2.0 (1.8, 2.0)	1.1 (−0.4, 1.5)	0.150
Over weight (z >1.45) (%)	17 (33)	11 (29)	3 (60)	3 (25)	
LV mass (z)	0.4 (−0.3, 1.1)	0.4 (−0.3, 1.0)^a,b^	1.2 (0.7, 1.6) ^a^	0.1 (−0.7, 0.6) ^b^	0.037
IVSd (z)	1.5 (0.3, 2.1)	1.3 (0.3, 1.8) ^a^	4.2 (2.6, 5.3)	1.3 (0, 2.0) ^a^	<0.001
LVPWd (z)	0.4 (−0.3, 1.2)	0.1 (−0.38, 0.6)	3.9 (3.8, 4.5)	1.0 (0.1, 1.8)	<0.001
E/A(z)	−1.3 (−1.6, −0.67)	−1.3 (−1.6, −0.7)	−1.6 (−1.7, −1.5)	−1.2 (−1.9, −0.4)	0.374
E/e’ (z)	1.3 (0.5, 2.18)	1.3 (0.3, 2.3)	1.3 (0.4, 2.2)	1.1 (0.7, 1.4)	0.315
RV S’ (z)	−1.1 (−1.7, −0.4)	−1.2 (−1.9, −0.3)	−0.4 (−0.8, −0.4)	−1.3 (−1.7, 1.0)	0.955
EF (%)	70.6 (67.5, 75.8)	69.8 (67.0, 75.3)	81.9 (73.0, 84.0)	71.6 (68.4, 73.7)	0.399
FS (%)	39.7 (37.1, 44.4)	38.6 (37.0, 44.2)	49.2 (41.0, 52.0)	40.5 (38.3, 41.4)	0.474
Strain (z)	1.4 (0.3, 2.1)	0.9 (0, 1.8)	1.8, (0.9, 2.)	2.0 (1.4, 2.4)	0.068
Uric acid (mg/dL)	6.0 (4.0, 7.9)	7.1 (5.7, 8.4)	3.9 (3.3, 4.2) ^a^	3.3 (2.7, 3.8)^a^	<0.001
Lactate (mmol/L)	2.2(1.8, 3.0)	2.5(2.0, 3.0)	1.7(1.3, 2.1) ^a^	1.7(1.2, 2.1) ^a^	<0.001
Total Col.(mg/dL)	190.0 (157.5, 228.5)	200.0 (173.0, 233.5) ^a^	190.0 (179.0, 216.0) ^a,b^	147.5 (135.8, 168.3)^b^	0.003
HDL (mg/dL)	43.0 (38.5, 55.0)	43.5 (38.3, 52.5) ^a,b^	39.0 (34.0, 40.0) ^a^	53.5 (42.5, 64.5) ^b^	0.023
LDL (mg/dL)	94.0 (79.0, 123.0)	107.5 (54.8, 129.5) ^a^	91.0 (79.0, 116.0) ^a,b^	83.0 (62.5, 92.8) ^b^	0.017
TG (mg/dL)	166.0(114.5, 281.5)	190.0(137.5, 371.5) ^a^	264.0(159.0, 378.0) ^a^	89(65.5, 110.0)	0.010
AST (U/L)	33.0(25.0, 60.0)	31.0(24.8, 43.0) ^a^	299.0(80.0, 372.0)	33.0(25.0, 35.2) ^a^	0.004
ALT (U/L)	25.0(17.0, 72.0)	26.5(17.8, 57.0) ^a^	367.0(100.0, 636.0)	17.0(15.0, 24.8) ^a^	0.001
CK (U/L)	122.0(95.0, 144.0)	120.0(90.0, 134.0) ^a^	574.0(259.5, 936.0)	130.0(99.8, 182.8) ^a^	0.009

*p* values represent overall differences across groups as determined by one-way analysis of variance (ANOVA) or Kruskal-Wallis’ H-test for continuous variables. ^a^,^b^ Same letters indicate no statistical significance based on Tukey’s post-hoc analysis or Bonferroni Correction Method. AST, aspartate transaminase; ALT, alanine transaminase; BMI, body mass index; CK, creatin kinase; EF, ejection fraction; FS, fractional shortening; LV, left ventricle; LVPWd, left ventricular posterior wall thickness diameter; IVSd, interventricular septum diameter; TG, triglyceride; Total Col., total cholesterol; z, z-score.

**Table 3 ijerph-20-02191-t003:** Correlation analysis of risk factors for LV mass z score in child age.

Variables	r	*p*-Value
Age (years)	0.002	0.998
BMI z-score	0.374	0.005
Gender	−0.167	0.194
Uric acid	0.039	0.776
Lactate	0.030	0.830
Total Col.	−0.017	0.903
TG	−0.072	0.601
HDL	−0.283	0.036
AST	0.235	0.084
ALT	0.228	0.094
CK	0.377	0.005

AST, aspartate transaminase; ALT, alanine transaminase; BMI, body mass index; CK, creatin kinase; TG, triglyceride; Total Col., total cholesterol.

**Table 4 ijerph-20-02191-t004:** Multiple linear regression for LV mass z score in child age.

Variables	Unstandardized B	Coefficients Std Error	t	*p*-Value
BMI z score	0.327	0.138	2.408	0.022
CK	0.002	0.001	2.364	0.020

BMI, body mass index; CK, creatin kinase.

**Table 5 ijerph-20-02191-t005:** Baseline characteristics and results of echocardiography in adult patients.

No.	type	Age (year)	gender	BMI	LMI (g/BSA)	IVSDd (mm)	LVPWd (mm)	E/A	E/e’	RV S’ (cm/s)	EF (%)	GLS (%)
1	1	20	M	21.7	104.1	11.1	10.7	1.93	8.3	16.0	71.4	−20.0
2	1	27	F	25.5	94.6	7.87	8.59	1.62	7.85	11.4	82.0	−19.1
3	1	30	F	26.4	58.9	7.16	7.16	1.45	10.90	12.0	68.6	−20.7
4	1	31	M	21.6	85.3	6.99	6.65	1.60	10.18	13.0	61	−19
5	1	34	M	20.4	109.4	7.65	6.99	0.78	11.38	13.0	74	−17.5
6	3	25	F	19.3	100.7	8.85	7.93	2.34	8.73	12.2	75.4	−21
7	3	36	M	22.5	72.19	9.67	9	1.36	6.10	10.0	56	−21

BMI, body mass index; EF, ejection fraction; GLS, global longitudinal strain; LMI, left ventricle mass index; LVPWd, left ventricular posterior wall thickness diameter; IVSd, interventricular septum diameter.

## Data Availability

The data used to support the findings of this study are available from the corresponding author upon request.

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
