# Peer review of "Echocardiographic Assessment of Patients with Glycogen Storage Disease in a Single Center"

_ijerph, 2023, doi:10.3390/ijerph20032191_

Round 1

Reviewer 1 Report

A paper of excellent quality, with a relevant theme. Some more of the writing needs to be revised, considering grammatical considerations, so I suggest corrections throughout the text.

Author Response

We thank the reviewer for reviewing our manuscript. We have addressed the comments and corrected grammatical errors and ambiguities.

Reviewer 2 Report

Congratulations on the study. It's a niche subject but the overall sample is quite large for a single center. The methods are very clearly described and the presentation is very readable. Particularly liked the illustrations. excellent discussion 

Author Response

We appreciate your systematic review of our study. We plan to enroll and study more for GSD patients in the future. Thanks again for the encouragement of our research.

Reviewer 3 Report

The biggest question is the whole statistical analysis in the materials & methods and results sections: in the main body text and in the tables 1 and 2. For example, the level of BMI was presented as 19.8 ± 1.8 and 17.6 ± 2.8 for the small number of patients that was not correct to do so.

Author Response

We thank the reviewer for pointing out this potential statistical error. We attempted to calculate the mean and standard deviation of continuous variables that follow normality; however, as pointed out, the number of patients with types III and IX GSD was small. Accordingly, the median and quartile were calculated instead, and Tables 1 and 2 were modified. For consistency, the median and quartile were also used for the group of more than 30 patients.

Reviewer 4 Report

The authors presented the echo characteristics of GSD patients, a rare metabolic disease. Some concerns I have:

- Authors included 62 patients in the analysis; however, most were type I patients. Then these patients were divided into pediatric and adult, decreasing the number of each GSD type. Therefore, for Tables 1 and 2, the analysis among different groups may have bias and not indicate meaningful results. Also, as most patients that authors had included in the study, GSD I may not have a cardiac influence; since the article's primary purpose is to evaluate how glycogen accumulation in the heart influences heart function, not to discuss the difference between different GSD types. Authors may combine all GSD patients together or just compare GSD 1 with GSD 3/6/9. In other words, the study design needs to be revised to better reflect the author's purpose.

- There has several similar papers to discuss the topic. For example, this paper (Echocardiographic Manifestations of Glycogen Storage Disease III: Increase in Wall Thickness and Left Ventricular Mass over Time. https://www.ncbi.nlm.nih.gov/pmc/articles/PMC3763918/) did similar work. Although this report only included 33 patients, they all were GSD type 3, with apparent cardiac involvement. To compare, in this manuscript, the authors only have 9 patients with type 3; therefore, it will compromise the author's conclusion and undermine the impression of cardiac change from the author's analysis.

Author Response

Response: We appreciate the comprehensive review of our manuscript by the reviewer. The comments have helped us improve our manuscript.

The authors presented the echo characteristics of GSD patients, a rare metabolic disease. Some concerns I have:

  1. Authors included 62 patients in the analysis; however, most were type I patients. Then these patients were divided into pediatric and adult, decreasing the number of each GSD type. Therefore, for Tables 1 and 2, the analysis among different groups may have bias and not indicate meaningful results. Also, as most patients that authors had included in the study, GSD I may not have a cardiac influence; since the article's primary purpose is to evaluate how glycogen accumulation in the heart influences heart function, not to discuss the difference between different GSD types. Authors may combine all GSD patients together or just compare GSD 1 with GSD 3/6/9. In other words, the study design needs to be revised to better reflect the author's purpose.

Response: We thank the reviewer for this insight. As pointed out, there were limitations when comparing the three groups since the sample size for each GSD type varied considerably. We have included this limitation in the discussion. However, the purpose of this study was not to compare the types of GSD, but to address the need for cardiac evaluation in patients with all types of GSD, including types which are not known to be have cardiac involvement. Therefore, we wanted to show the characteristics of each type in Tables 1 and 2. We are conducting further research to produce comparable data in more patients with types III and IX GSD.

  1. There has several similar papers to discuss the topic. For example, this paper (Echocardiographic Manifestations of Glycogen Storage Disease III: Increase in Wall Thickness and Left Ventricular Mass over Time. https://www.ncbi.nlm.nih.gov/pmc/articles/PMC3763918/) did similar work. Although this report only included 33 patients, they all were GSD type 3, with apparent cardiac involvement. To compare, in this manuscript, the authors only have 9 patients with type 3; therefore, it will compromise the author's conclusion and undermine the impression of cardiac change from the author's analysis.

Response: As pointed out, the small number of patients with GSD type III included in our study may underestimate the effects of GSD on the heart. Our primary objective was to highlight the need for cardiac evaluation in patients with a type of GSD where cardiac structure and function evaluations are typically disregarded. We acknowledge that this needs to be expressed more clearly; therefore, the conclusion has been revised as follows.

“In our study, metabolic abnormalities, such as hyperuricemia, lactic acidosis, and dyslipidemia, caused by GSD did not have a significant effect on the function of the heart; however,, there was a significant change in the myocardial thickness with weight gain. Given that patients with GSD tend to be overweight or obese, steady cardiac evaluation through echocardiography is required for all types of GSD, including types not known to be associated with cardiac involvement. Not only myocardial thickness, but also the evaluation of diastolic function and 2D STE will enable early detection of subclinical cardiac dysfunction. Future studies with a larger number of adult patients will be able to observe the long-term effects of metabolic abnormalities on the hearts of patients with GSD. In addition, it is necessary to find out whether diet, optimal metabolic control, and exercise management in patients with GSD have a positive effect on changes in cardiac structure and function through follow-up echocardiography.” (Line 357-369)

Round 2

Reviewer 4 Report

Thank the authors for the revised version of the manuscript. The manuscript has a more clear purpose than before, which focuses on the cardiac consequence of the metabolic status change caused by GSD, and it makes the study reasonable.  I believe it is much better than the previous version.